# Bisphenol AF Induces Prostatic Dorsal Lobe Hyperplasia in Rats through Activation of the NF-κB Signaling Pathway

**DOI:** 10.3390/ijms242216221

**Published:** 2023-11-12

**Authors:** Sisi Huang, Kaiyue Wang, Dongyan Huang, Xin Su, Rongfu Yang, Congcong Shao, Juan Jiang, Jianhui Wu

**Affiliations:** 1Shanghai Engineering Research Center of Reproductive Health Drug and Devices, NHC Key Lab of Reproduction Regulation, Shanghai Institute for Biomedical and Pharmaceutical Technologies, Pharmacy School, Fudan University, Shanghai 200237, China; sisihuang22@m.fudan.edu.cn (S.H.); 17863032586@163.com (K.W.); hdy043@163.com (D.H.); suxiaoxin1982@163.com (X.S.); yangrongfu82@163.com (R.Y.); shaocongcongscc@163.com (C.S.); jiangjuan1106@163.com (J.J.); 2Shanghai-MOST Key Laboratory of Health and Disease Genomics, NHC Key Lab of Reproduction Regulation, Shanghai Institute for Biomedical and Pharmaceutical Technologies, Pharmacy School, Fudan University, Shanghai 200237, China

**Keywords:** bisphenol AF, prostatic hyperplasia, nuclear transcription factor-κB, cyclooxygenase-2

## Abstract

Bisphenol AF (BPAF) represents a common environmental estrogenic compound renowned for its capacity to induce endocrine disruptions. Notably, BPAF exhibits an enhanced binding affinity to estrogen receptors, which may have more potent estrogenic activity compared with its precursor bisphenol A (BPA). Notwithstanding, the existing studies on BPAF-induced prostate toxicity remain limited, with related toxicological research residing in the preliminary stage. Our previous studies have confirmed the role of BPAF in the induction of ventral prostatic hyperplasia, but its role in the dorsal lobe is not clear. In this study, BPAF (10, 90 μg/kg) and the inhibitor of nuclear transcription factor-κB (NF-κB), pyrrolidinedithiocarbamate (PDTC, 100 mg/kg), were administered intragastrically in rats for four weeks. Through comprehensive anatomical and pathological observations, as well as the assessment of PCNA over-expression, we asserted that BPAF at lower doses may foster dorsal prostatic hyperplasia in rats. The results of IHC and ELISA indicated that BPAF induced hyperplastic responses in the dorsal lobe of the prostate by interfering with a series of biomarkers in NF-κB signaling pathways, containing NF-κB p65, COX-2, TNF-α, and EGFR. These findings confirm the toxic effect of BPAF on prostate health and emphasize the potential corresponding mechanisms.

## 1. Introduction

An endocrine-disrupting chemical (EDC) is defined as “an exogenous chemical, or mixture of chemicals, that interferes with any aspect of hormone action”, which may be found in various everyday commodities, including plastic bottles, food, drugs, toys, agricultural products, and cosmetics [1,2]. Recently, EDCs have received widespread attention because of their potential to interfere with the physiological regulation of hormones crucial for the growth and development of reproductive tissues, even at low levels [3,4,5]. Notably, these exogenous chemicals interfere with the binding of hormones to their corresponding receptors, such as the estrogen receptor (ER) and androgen receptor (AR), resulting in activation or antagonistic effects [6]. It is well known that androgens control the normal growth and functional activity of the prostate [7]. Simultaneously, estrogens also play an important role in male-sex-hormone secretion, growth, differentiation, and homeostasis of normal and malignant prostate cells [8]. In aging males, the decline in testosterone levels results in a conspicuous elevation in the estrogen-to-androgen ratio [9]. The relative increase in estrogen level has been observed to stimulate excessive prostate cell proliferation, culminating in hyperplasia [10]. Furthermore, reports indicate that EDCs, resembling the action of estrogen, have the capacity to disrupt the reprogramming of prostate stem cells, thereby exerting an influence on prostate health [7].

Bisphenol A (BPA), a common environmental estrogen with an endocrine-disrupting effect, exerts its effects through mimicking endogenous estrogenic function within the human body. This mimetic behavior leads to a disruption of endocrine balance by affecting the synthesis and metabolism of hormones, ultimately culminating in the development of severe prostate pathologies, including benign prostatic hyperplasia (BPH) and prostate cancer [7,11]. Over the past two decades, regulatory measures have been implemented to restrict BPA usage in children’s commodities and thermal paper receipts. Consequently, novel BPA analogues, such as bisphenol S (BPS), bisphenol F (BPF), and bisphenol AF (BPAF), each sharing structural and functional similarities with BPA, have emerged and gained utilization. Among them, BPAF, also recognized as hexafluorobisphenol A, is produced in substantial quantities for the formulation of organic synthetic materials, including resins, fluoropolymers, and plastics. Skin contact is the primary route of exposure [12]. Recent studies have found that exposure to higher concentrations of BPAF, exceeding environmental levels or arising from occupational contexts, results in pronounced gonadotropin suppression and induces Leydig cell apoptosis or autophagy [13]. Other perspectives have been proposed that BPAF may variably damage the pituitary–gonad-axis function by reducing serum testosterone levels in rat models, ultimately leading to increased testicular weight [14]. However, the existing body of research addressing the prostate-specific toxicological implications of BPAF remains limited.

BPAF is recognized for its heightened affinity and enhanced estrogenic potency in comparison with BPA [15]. This is attributed to the fact that the trifluoromethyl group (CF_3_) in the BPAF structure has higher electronegativity than the methyl group (CH_3_) in BPA. Therefore, exposure to BPAF via various routes, such as skin contact, inhalation, or dietary intake, has the potential to induce comparable or more potent toxic effects on the growth and development of the prostate, but this conjecture remains to be further verified. Since BPAF could act both as an activator of ERα and by antagonizing ERβ [16], this dual modality of action is of great significance for exploring the interference pathway of the estrogen signal. It is posited that BPAF has the capacity to not only disrupt hormone homeostasis in vivo but also to augment susceptibility to prostate lesions. Therefore, the mechanisms underlying prostate toxicity induced by BPAF require further exploration.

As the core member of inflammatory mediators, the abnormal activation of the NF-κB signaling pathway plays a pivotal role in governing the intricate processes of proliferation and apoptosis in prostate cells. This regulation is achieved primarily through its mediation of the inflammatory response and modulation of hormone receptor expression within the prostate tissue. The NF-κB/Rel family includes five transcription factors, NF-κB1 (p50), NF-κB2 (p52), c-Rel, RelA (p65), and RelB (p66), all sharing a common Rel homology domain (RHD) [17]. In the resting state, the NF-κB subunit is trapped in the cytoplasm through the IκB protein and unprocessed p105 and p100 proteins [18,19]. The IκB kinase (IKK) complex, comprising distinct subunits including catalytic subunits (IKKα and IKKβ) and NF-κB essential regulatory factors, is central to this regulatory framework. Upon stimulation by interleukin-1β, tumor necrosis factor and bacterial-derived ligands, it provokes phosphorylated degradation of IκB-α and the release of NF-κB. This results in nuclear translocation of homodimers or heterodimers of NF-κB subunits, thereby activating or inhibiting gene transcription [18,20]. Among these configurations, the heterodimer composed of NF-κB p65 and NF-κB p50 is the most important binding form [21]. At the same time, NF-κB p65 has been proven to be closely related to the epidermal growth factor receptor (EGFR), tumor necrosis factor α (TNF-α), prostaglandin E2 (PGE2)—the main metabolite of the arachidonic acid pathway catalyzed by COX-2, and COX-2 itself [22,23,24]. Therefore, NF-κB p65 has gradually emerged as a potential biomarker of inflammatory diseases and various tumors. Through in vivo screening and verification, our study found that low doses of BPAF within the range of 10 to 90 μg/kg promotes ventral prostatic hyperplasia in rats, which is mainly achieved through the activation of the signal transduction pathways of NF-κB p65, COX-2, TNF-α, and EGFR [11]. However, the role and mechanism of BPAF in the dorsal lobe of the prostate (DLP) of rats remains to be confirmed.

In this study, 50 male rats were used to investigate the effects of low-dose BPAF (10–90 μg/kg) exposure on the development of DLP hyperplasia, as well as the expression of NF-κB and its related biomarkers in DLP of rats, including NF-κB p65, COX-2, TNF-α, and EGFR. Therefore, we fully confirmed the role of BPAF in the induction of BPH and comprehensively verified the expression relationship between NF-κB and its downstream regulatory factors in this process, which provided an important reference for determining the prostatic toxicity of BPAF and developing targeted therapeutic strategies for BPH.

## 2. Results

### 2.1. Anatomical Analysis of Prostates in Rats in Response to BPAF and PDTC

Four weeks after oral administration of BPAF and the NF-κB inhibitor pyrrolidinedithiocarbamate (PDTC), anatomical data showed that the prostate organ coefficient (Figure 1A) and prostate–brain coefficient (Figure 1B) in the BPAF (90 μg/kg) group increased significantly, reflecting the prostate weight gain induced by BPAF. Combined administration of inhibitors could significantly reduce the prostate organ coefficient and prostate–brain coefficient, showing the inhibitory effect of the NF-κB-signaling-pathway inhibitor on the promotion of prostate weight by BPAF.

### 2.2. Histopathological Analysis of Prostates in Rats Induced by BPAF

The histological examination of DLP in rats has revealed noteworthy pathological manifestations. Specifically, administration of 10 and 90 μg/kg of BPAF in isolation has been associated with abnormal thickening of prostate epithelium, increased number and dense distribution of glands, acinar deformation, and other morphological changes. Notably, the histopathological transformations observed in the prostate as induced by BPAF were observed to be significantly alleviated following combined administration of inhibitory agents (Figure 1C). Moreover, the measurement of the epithelial height of DLP in rats (Figure 1E) showed that 10 and 90 μg/kg of BPAF demonstrated a propensity to elevate the epithelial height of DLP. At the same time, the epithelial height of DLP in the BPAF (10 μg/kg) + PDTC group and BPAF (90 μg/kg) + PDTC group was observed to be significantly reduced in comparison with the corresponding single-administration groups.

### 2.3. Analysis of PCNA Expression in Rat Prostate Tissue Induced by BPAF

The PCNA protein is mainly expressed in the nucleus, and its positive expression rate stands as a pivotal index for evaluating cell proliferation within the rat prostate tissue. The results of IHC staining showed that the nuclear positive for PCNA could be stained brown and expressed in the epithelium of DLP in all groups (Figure 1D). Compared with the vehicle group, the administration of 10 μg/kg and 90 μg/kg doses of BPAF elicits a significant augmentation in the positive expression rate of PCNA in DLP (Figure 1F). After administration of the corresponding inhibitors, the expression of PCNA in DLP showed a significant downward trend.

### 2.4. Localization and Qualification of Key Regulators of NF-κB Signaling Pathway

The immunohistochemical assessment of NF-κB p65, COX-2, TNF-α, and EGFR in DLP of rat prostate was conducted to illustrate the cellular localization and expression levels of these key biomarkers. The identification of a brown-hued staining pattern in the tissue sections denoted positive expression. As previously mentioned, NF-κB is mainly located in the cytoplasm in resting cells. Upon cellular activation, the phosphorylation of IκB by IKK results in its degradation, leading to the translocation of NF-κB into the nucleus and binding to the target gene to activate related gene transcription. COX-2, TNF-α, and EGFR are central regulators of the NF-κB signaling pathway, which are mainly expressed in the cytoplasm of the epithelial tissue of DLP (Figure 2).

The results of the semi-quantitative analysis revealed that BPAF could significantly enhance the positive expression of NF-κB p65, COX-2, and EGFR in DLP (Figure 3A,B,D) but demonstrated no significant effect on the expression of TNF-α (Figure 3C). Furthermore, 90 μg/kg of BPAF combined with the inhibitor PDTC exhibited a notable reduction in the positive expression of NF-κB p65, COX-2, and EGFR in DLP. However, the mitigating influence of PDTC on the expression of TNF-α was not deemed statistically significant.

### 2.5. BPAF Stimulated the Expression of NF-κB p65, COX-2, TNF-α, and EGFR

The results of the tissue-based enzyme-linked immunosorbent assay (ELISA) reached a similar conclusion (Figure 3E–H). Exposure to BPAF in isolation increased the levels of NF-κB p65, COX-2, TNF-α, and EGFR, but the changes of COX-2 (Figure 3F) and TNF-α (Figure 3G) in DLP of each group exhibited comparatively modest deviations from baseline levels. Among these groups, 90 μg/kg of BPAF induced the most significant disturbance on the expression of NF-κB p65 (Figure 3E). 

Subsequent intervention with PDTC resulted in the downregulation of NF-κB p65, COX-2, TNF-α, and EGFR expressions within DLP, with a particularly conspicuous inhibitory effect on EGFR expression (Figure 3H). Furthermore, it is noteworthy that neither sole exposure to BPAF nor its combination with PDTC exerted a substantial impact on TNF-α expression in DLP, aligning with the observations from the IHC conclusion (Figure 3G).

## 3. Discussion

Owing to the widespread utilization of BPA and its analogues, many people around the world have been exposed to bisphenols. BPAF, in particular, is distinguished by its capacity to elicit both genomic and non-genomic effects through the activation of nuclear ER and G-protein-coupled-estrogen-receptor (GPER) pathways, respectively, while concurrently manifesting heightened endocrine activity compared with BPA [9]. Recent studies have shown that exposure to BPAF engenders neurobehavioral damage patterns [25], cardiovascular toxicity [26], and reproductive and developmental toxicity [27], which exhibit similarities to the effects associated with BPA. Therefore, our research endeavors are directed towards exploring the impact of environmentally relevant doses of BPAF exposure on the induction of hyperplasia within DLP in rat models, while concurrently scrutinizing the modulatory effects of NF-κB-signaling-pathway inhibition on the interference phenomenon.

It has been documented that the exposure to bisphenols, such as BPA, BPS, and BPF, for two months has resulted in an increase in the weights of both the bladder and prostate [28], which was similar to the evaluation of BPAF promoting prostate weight in our study. Specifically, the administration of 90 μg/kg of BPAF resulted in a statistically significant elevation in both the prostate organ coefficient and the prostate–brain coefficient. At the same time, our histomorphological analysis of prostates in rats revealed several distinctive alterations, containing an increasing trend in the epithelial height of DLP, pronounced deformation and extrusion of glandular lumens, and a concomitant reduction in the dimensions of glandular cavities. These findings collectively suggested that exposure to BPAF disrupted the normal growth and development of the prostate gland. To further verify the above hypothesis, we performed a comprehensive assessment involving the localization detection and semi-quantitative analysis of PCNA in DLP. As an important index to evaluate the effect of bisphenols on cell proliferation, we found that the positive expression rate of proliferating cell nuclear antigen increased after the treatment of BPAF, indicating the potential of inducing prostate proliferation in rats. Therefore, we propose that BPAF could promote abnormal hyperplasia of the prostate. Additionally, combined administration of NF-κB-signaling-pathway inhibitors exhibited a mitigating effect on the above evaluation indexes, suggesting that the NF-κB-signaling-pathway inhibitors may confer a capacity to alleviate the prostate toxicity induced by BPAF.

In addition to exerting influences on estrogen and androgen receptor signaling pathways, bisphenols also exhibit a pronounced inflammatory effect, which is closely related to their capacity to promote cell proliferation and tissue proliferation [29,30,31]. Consequently, within the context of exploring the relevant molecular mechanisms, the convergence of diverse inflammatory reactions, particularly the NF-κB and its downstream signaling pathways, has gradually emerged as a focal point of research attention [32,33,34]. In human prostate epithelium, the inflammasome component NLRP3 plays a pivotal role in regulating immune responses and maintaining homeostasis. The activation of NLRP3 depends on the regulation of NF-κB and the generation of ROS, thereby facilitating the release of an array of pro-inflammatory factors that trigger a cascading inflammatory response [35]. In this study, crucial indexes of the NF-κB signaling pathway in DLP of rats were spatially localized and quantitatively detected. It is concluded that BPAF is capable of inducing BPH by affecting the activity of the NF-κB signaling pathway; the key results of anatomical, histological, IHC, and ELISA analyses are shown in Table 1. Specifically, BPAF is demonstrated to have the tendency to upregulate the expression of NF-κB p65, COX-2, TNF-α, and EGFR in DLP, which appears to be a key target contributing to prostatic hyperplasia triggered by BPAF exposure. These findings collectively offer a convincing explanation for the mechanism of BPAF-induced cell proliferation by synthesizing the relationship among hormonal signaling pathways, inflammatory responses, and oxidative stress.

Abnormal activation of NF-κB p65 and the upregulation of COX-2 are commonly observed phenomena in various forms of prostate lesions. Examination of the COX-2 gene sequence has revealed the presence of two κB sites located at its 5’ terminus [36], thereby implicating NF-κB in the modulation of COX-2 gene transcription. For instance, in a rat model induced with BPH via testosterone propionate, the hyperphosphorylation of NF-κB p65 leads to the elevation of pro-inflammatory mediators, such as IL-1β, IL-6, TNF-α, and COX-2, while also increasing the expression of Bcl2/Bax, consequently contributing to the occurrence and development of BPH in rats [37]. In this study, we found that low-dose exposure to BPAF could promote the expression of NF-κB p65 and COX-2 in DLP of rats. After inhibiting the activity of the NF-κB signaling pathway, the expression of NF-κB p65 and COX-2 were downregulated. This outcome parallels the findings reported by Xie et al. in their investigation of BPH [38], which reveals an age-related decline in the expression of cystic-fibrosis transmembrane conductance regulator (CFTR) in rat prostate tissue, a factor that negatively regulates the NF-κB/COX-2/PGE2 signaling pathway in prostate epithelial cells, thereby stimulating its activity. This revelation opens up a novel avenue for potential therapeutic interventions in the context of BPH.

TNF-α is also a common inflammatory cytokine, capable of inducing IκB phosphorylation, ubiquitination, and subsequent nuclear translocation of NF-κB, thereby mediating the survival and apoptosis of normal cells. This differential responsiveness to TNF-α-induced biological reactions between human normal prostate epithelial cells and prostate cancer cells stems from this fundamental molecular mechanism [39]. Recent studies have shown that sustained exposure to BPA is associated with increased gene expression levels of pro-inflammatory markers, such as TNF-α, COX-2, and the pro-oxidative factor p53 within the cerebral tissue of rats, thus aggravating cerebral inflammation and neurological dysfunction [40]. Our previous study found that BPAF exerts a stimulatory influence on the excessive production of TNF-α in the ventral lobe of the prostates in rats [11]. In the present study, we have discerned that the positive staining intensity and tissue content of TNF-α in DLP have not exhibited a statistically significant disparity, which corresponds to the results of the ELISA detection of NF-κB p65. Therefore, it is speculated that there presents a regulatory interplay involving TNF-α and NF-κB in the prostates of rats, but the regulatory effect exerted by TNF-α may not be notably influential in the context of low-dose BPAF-induced hyperplasia in DLP. On the contrary, TNF-α showed significant changes in the ventral lobe of the prostate [11]. The regulation of TNF-α expression is intricately influenced by a multitude of variables, including intricate interactions between tissue-specific factors and signaling pathways within distinct anatomical regions. We further posit that the difference in TNF-α expression levels observed between the dorsal and ventral lobes of the prostate may be ascribed to inherent structural disparities between these two anatomical regions. 

EGFR is a transmembrane protein and belongs to the tyrosine kinase receptor, which can be activated by epidermal growth factor (EGF), transforming growth factor α (TGF-α), and integrin. As a mitogenic factor in keratinocytes and fibroblasts, activation of EGFR could stimulate the process of epithelial–mesenchymal transition (EMT) and trigger the NF-κB signaling pathway in prostate epithelial cells, thus triggering the epithelial cell survival process and protecting cells from apoptosis [41]. Furthermore, EGFR also functions as an upstream regulator, influencing the activity of the Akt/NF-κB signaling pathway in prostate cancer cells [42]. Our previous studies have shown that the upregulation of the EGF/EGFR signaling pathway, mediated by both estrogen and androgen, plays an important role in the promotion of DLP induced by BPA exposure in rats [43]. The present study is specifically focused on assessing the effect of BPAF on the expression of EGFR and establishing a connection between NF-κB and EGFR in the context of BPH induced by BPAF. Our findings demonstrate that BPAF has the capacity to upregulate the expression of EGFR in the prostate. Furthermore, upon inhibiting the activity of the NF-κB signaling pathway, the abnormal activation of EGFR was also weakened, suggesting that EGFR may play a role in regulating pathological hyperplasia of the prostate as a downstream modulator of NF-κB.

In conclusion, our study assumes a pivotal role in complementing the existing body of knowledge pertaining to reproductive toxicity and the mechanisms underlying the impact of low-dose BPAF exposure. The findings and insights derived from our research are poised to enhance public awareness concerning the intimate relationship that exists between environmental exogenous estrogens, such as BPAF, and the etiology of prostate diseases. This, in turn, holds the potential to make substantial contributions towards the formulation of strategies for the prevention and treatment of benign prostatic hyperplasia, which may be attributed to the exposure to bisphenols. Our work also serves the vital function of providing an early warning system for the monitoring and management of BPA and its analogues, thereby supporting efforts aimed at safeguarding public health and environmental well-being. However, it remains imperative to delve further into the intricacies of how BPAF affects the mechanism of NF-κB-signaling-pathway transduction, and whether there are other mechanisms needs to be further studied. Moreover, future research might broaden the scope of our investigations beyond animal and cellular models in the study of prostate disease. For example, incorporating the development of prostate organ models may serve as a valuable avenue for elucidating the interference effects and underlying mechanisms associated with environmental estrogens.

## 4. Materials and Methods

### 4.1. Animal Treatment

Fifty specific-pathogen-free (SPF) male Sprague–Dawley (SD) rats, within the weight range spanning from 370 to 400 g and the age range of 77 to 83 days, were purchased from Zhejiang Vital River Laboratory Animal Technology Co., Ltd. (Jiaxing, China). Under the light/dark cycle of 12 h:12 h, all animals were raised and drunk freely (Shanghai Shilin Science & Tech Co., Ltd., Shanghai, China) within an environment maintained at a temperature range of 20–26 °C and humidity levels between 40–70%. All rats were treated in accordance with the Guidelines for the Care and Use of Laboratory Animals.

After a period of 5 days devoted to acclimatization, it was observed that all the rats exhibited a state of optimal health. According to their body weight, the rats were randomly divided into five groups (*n* = 10): the vehicle, BPAF (10 μg/kg), BPAF (90 μg/kg), BPAF (10 μg/kg) + PDTC, and BPAF (90 μg/kg) + PDTC. The administration of test substances was conducted through intragastric gavage, utilizing a 5% sodium carboxymethyl cellulose (CMC-Na) solution as the solvent. Specifically, BPAF was administered at doses of 10 μg/kg and 90 μg/kg, while PDTC was administered at a dose of 100 mg/kg to the respective rat groups over a duration of four weeks. After administration, the rats were humanely euthanized by exposure to carbon dioxide asphyxiation. Subsequently, the brain and ventral lobe and dorsal lobe of the prostate were dissected and weighed, and the total weight of the prostate was calculated. Part of the tissue was fixed in paraformaldehyde solution and used for histopathological examination, and the other part was frozen in liquid nitrogen at −80 °C for protein detection. The assessment of prostate weight gain among the rats was undertaken through the computation of two distinct ratios: the ratio of 1000 times the total prostate weight to the terminal body weight and the ratio of 100 times the total prostate weight to the terminal brain weight.

### 4.2. Histopathological Assessment

The prostate tissue fixed by paraformaldehyde for 48 h was removed and trimmed, dehydrated, cleaned, waxed, and embedded in paraffin, and every paraffin-embedded tissue was cut into 4 μm thick slices with a microtome (Leica, Shanghai, China). The deparaffinization process involved the use of xylene, followed by sequential immersion in 100% ethanol to achieve rehydration, transitioning to 75% ethanol. Hematoxylin and eosin (H&E) staining was performed with a staining duration of 3–5 min and sealed with neutral gum. Six slices of each group were observed and photographed under an inverted microscope (Olympus, Tokyo, Japan). The epithelial height of each tissue sample was measured 10 times using image analysis software (Olympus CKX53, Tokyo, Japan).

### 4.3. IHC Analysis

The dewaxing and rehydration procedures were conducted in accordance with the established H&E staining protocol in the initial stage of experimentation. Subsequently, tissue specimens were immersed in a 0.01 M citrate buffer solution (pH 6.0) and subjected to a 20 min microwave heating step for antigen retrieval. To mitigate endogenous peroxidases activity and minimize non-specific binding, oxidase blocking solution and normal nonimmune serum were employed. The histological sections were incubated using primary antibodies against PCNA (1:100, AF1363, Beyotime, Shanghai, China), p-NF-κB p65 (1:75, AF5881, Beyotime, Shanghai, China), COX-2 (1:75, 31296I11P56, Boster, Wuhan, China), TNF-α (1:250, AF8208, Beyotime, Shanghai, China), and EGFR (1:120, AF5153, Beyotime, Shanghai, China), and the incubation was carried out at 4 °C overnight. The histological sections were treated with secondary antibodies at 37 °C for 10 min. Catalysis was achieved using *Streptomyces* antibiotic peroxidase solution, followed by 3,3’-diaminobenzidine (DAB) and hematoxylin staining. The stained sections were then dehydrated, washed, and sealed with neutral gum. Eight images were captured from DLP of each group. To assess the level of positive expression, the PCNA and NF-κB p65, predominantly localized in the nucleus, were quantified by determining the ratio of positively stained nucleus to the total nucleus. At the same time, the semi-quantitative analysis of COX-2, TNF-α, and EGFR expression levels was conducted using the IHC profiler plugin of the ImageJ 1.8.0 software.

### 4.4. Tissue-Based ELISA

The frozen tissue was sectioned into smaller pieces, and their respective weights were accurately recorded. Subsequently, these tissue pieces were subjected to homogenization using phosphate-buffered saline (PBS) at a specified mass-to-volume ratio of 1:10. Following the homogenization step, the resulting mixture underwent centrifugation to facilitate the formation of a homogenate. The quantification of key protein markers, including rat NF-κB p65 (ab176648, Abcam, Cambridge, UK), COX-2 (CSB-EL13399r, CUSABIO, Wuhan, China), TNF-α (EK0526, Boster, Wuhan, China), and EGFR (ELR-RGFR-1, RayBiotech, Norcross, GA, USA), was performed in strict accordance with the manufacturers’ instructions provided with the respective ELISA kits. The assessment of protein content was carried out using a microplate reader (Tecan, Männedorf, Switzerland) and i-control^TM^ No.: 2.0 software.

### 4.5. Statistical Analysis

All the test data were expressed in the form of mean ± standard deviation (means ± SD). Statistical analyses were conducted employing IBM SPSS Statistics version 26.0 software (SPSS Inc., Chicago, IL, USA). Prior to any inferential testing, assessments for the normality of data distribution and homogeneity of variances were carried out, ensuring compliance with the requisite assumptions. Subsequently, a one-way analysis of variance (one-way ANOVA) was used to analyze the differences among groups, and pairwise comparisons were made by the least-significant-difference (LSD) method if the difference between groups was statistically significant. Finally, the outcomes of these statistical analyses were graphically displayed by using GraphPad Prism 9 software (GraphPad Software, San Diego, CA, USA). A value for *p* < 0.05 was considered to indicate a statistically significant difference.

## 5. Conclusions

In summary, our study revealed that exposure to BPAF at doses of 10 and 90 μg/kg has the capacity to induce pathological hyperplasia in DLP. Meanwhile, this exposure leads to an upregulation tendency of the expression of NF-κB p65, COX-2, TNF-α, and EGFR in the prostate tissue. It is noteworthy that the inhibition of the NF-κB signaling pathway’s activity results in a mitigated manifestation of aberrations in hyperplasia-related evaluation indexes, thus potentially counteracting the abnormal prostatic dorsal lobe hyperplasia induced by BPAF in rat models. It is imperative to acknowledge the intricate and diverse interplay of upstream and downstream effect factors in the NF-κB signaling pathway, thus highlighting the necessity for further exploration of the abundant gene products subject to NF-κB regulation in the context of BPAF-induced prostate toxicity.

## Figures and Tables

**Figure 1 ijms-24-16221-f001:**
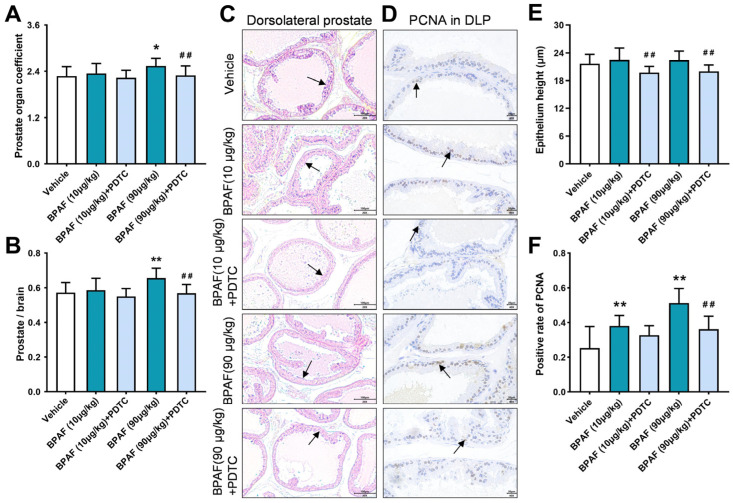
Weight gain and pathological changes of prostates in rats. Effect of exposure to bisphenol AF (BPAF) and the inhibitor of nuclear transcription factor-κB (NF-κB), pyrrolidinedithiocarbamate (PDTC), for 4 weeks on the prostate organ coefficient (**A**) and prostate–brain coefficient (**B**), *n* = 10. (**C**) The pathological changes of the tissues (200×, scale bar = 100 µm) and the prostate epithelium are pointed to with arrows. (**D**) The immunohistochemical images of proliferating cell nuclear antigen (PCNA) (400×, scale bar = 20 µm). Arrows indicate the positive expression. (**E**) The height of prostate epithelium of ventral prostates, *n* = 6. (**F**) Immunohistochemistry (IHC) quantification of PCNA is presented as the ratio of the positively expressed nucleus to the total nucleus, *n* = 8. Results are presented as means ± SD, analyzed using ANOVA, followed by LSD post hoc test. Comparison of the individual-administration group and the control: * *p* < 0.05, ** *p* < 0.01; comparison of the combined-administration group and the individual-administration group: ## *p* < 0.01. Prostate organ coefficient = 1000 × total prostate weight/terminal body weight. Prostate–brain coefficient = 100 × total prostate weight/terminal brain weight.

**Figure 2 ijms-24-16221-f002:**
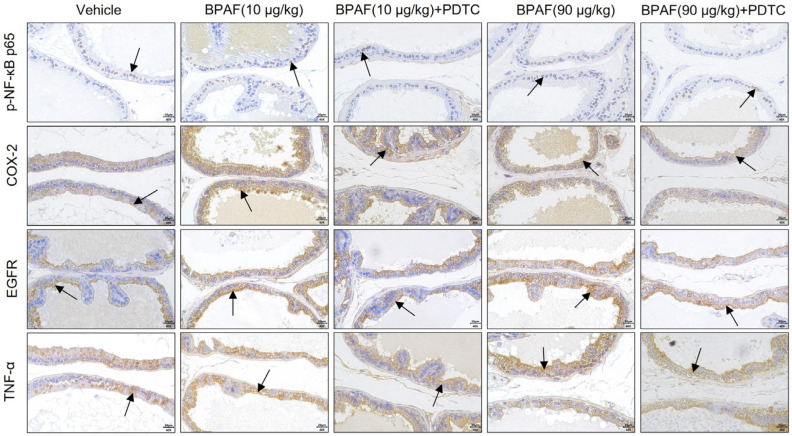
Localization observation of phosphorylated NF-κB p65 (p-NF-κB p65), cyclooxygenase-2 (COX-2), tumor necrosis factor-α (TNF-α), and epidermal-growth-factor-receptor (EGFR) expression in DLP. Immunohistochemical images of p-NF-κB p65, COX-2, TNF-α, and EGFR of the prostate tissues in the rats treated using BPA, BPAF, and PDTC (400×, scale bar = 20 µm). Arrows indicate positive expression.

**Figure 3 ijms-24-16221-f003:**
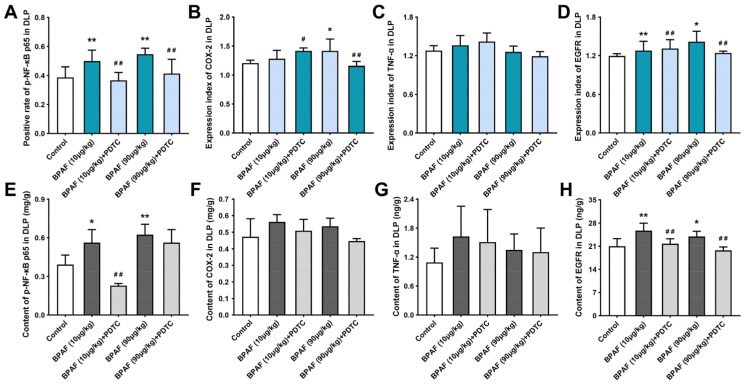
BPAF and PDTC induced the quantitative changes of p-NF-κB p65, COX-2, TNF-α, and EGFR in DLP. Effect of exposure to BPAF and PDTC on the rate of positive expression of p-NF-κB p65 (**A**); the semiquantitative expression levels of COX-2 (**B**), TNF-α (**C**), and EGFR (**D**) in the dorsal lobe of the prostate (DLP). Results are presented as means ± SD, analyzed using ANOVA, followed by LSD post hoc test, *n* = 8. Contents of NF-κB p65 (**E**), COX-2 (**F**), TNF-α (**G**), and EGFR (**H**) were obtained using enzyme-linked immunosorbent assay (ELISA). Results are presented as means ± SD, analyzed using ANOVA, followed by LSD post hoc test, *n* = 4. Comparison of the individual-administration group and the control: * *p* < 0.05, ** *p* < 0.01; comparison of the combined-administration group and the individual-administration group: # *p* < 0.05, ## *p* < 0.01.

**Table 1 ijms-24-16221-t001:** Key results of anatomical, histological, IHC, and ELISA analyses.

Group	Anatomical	Histological	IHC	ELISA
BPAF (10 μg/kg)	-	-	PCNA ↑, NF-κB p65 ↑, EGFR ↑	NF-κB p65 ↑, EGFR ↑
BPAF (10 μg/kg) + PDTC	-	Epithelial height of DLP ↓	NF-κB p65 ↓, COX-2 ↓, EGFR ↓	NF-κB p65 ↓, EGFR ↓
BPAF (90 μg/kg)	Prostate organ coefficient ↑^a^, prostate–brain coefficient ↑	-	PCNA ↑, NF-κB p65 ↑, COX-2 ↑, EGFR ↑	NF-κB p65 ↑, EGFR ↑
BPAF (90 μg/kg) + PDTC	Prostate organ coefficient ↓^b^, prostate–brain coefficient ↓	Epithelial height of DLP ↓	PCNA ↓, NF-κB p65 ↓, COX-2 ↓, EGFR ↓	EGFR ↓

^a^ Upward arrow (↑): Indicates a significant upregulation of the corresponding parameter following the treatment of the group. ^b^ Downward arrow (↓): Indicates a significant downregulation of the corresponding parameter following the treatment of the group.

## Data Availability

Data are available upon request.

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
