# Peer review of "Bisphenol AF Induces Prostatic Dorsal Lobe Hyperplasia in Rats through Activation of the NF-κB Signaling Pathway"

_ijms, 2023, doi:10.3390/ijms242216221_

Round 1

Reviewer 1 Report

Comments and Suggestions for Authors
  1. Provide more details on the doses selected and rationale for choosing them. Were they environmentally relevant? How do they compare to human exposure levels?
  2. Include a table summarizing the key results of anatomical, histological, IHC, and ELISA analyses for easier comparison across groups.
  3. Discuss any limitations of the study - e.g. lack of mechanistic insight into how BPAF affects NF-kB signaling.
  4. Provide more interpretation of the TNF-α results which did not show changes - why might this be?
  5. Discuss future directions and any clinical implications of the findings.
  6.  
  7.  
Comments on the Quality of English Language

The quality of English is good. Minor edits needed.

Author Response

Dear reviewer:

We sincerely thank you for your positive comments and constructive suggestions, which really help us a lot to improve our manuscript.

In Line with the reviewer’s comments, we extensively revised the manuscript. We described the changes in “Response” as follows. Accordingly, significant improvement was made in the manuscript to address these points. The changes made in the revised manuscript are highlighted in red.

We are grateful for your reconsideration of our manuscript, and we look forward to receiving comments from the reviewers.

Yours sincerely,

Jianhui Wu  

Reviewer 2 Report

Comments and Suggestions for Authors

The authors present their findings in support of the claim that BPAF results in an increase in prostate weight which is regulated via NFkB pathway. They demonstrate moderate increase in prostate organ coefficient following BPAF exposure which gets reverted when combined with treatment with NFkB inhibitor (PDTC). They also report an increase in height of prostate epithelium (not significant) and increase in PCNA post exposure to BPAF. They further supported their claims with IHC and ELISA data, most of which also demonstrated either no significance or moderate changes between the groups of interest. The article can benefit from additional experiments and deeper analysis of current IHC data. Some of the comments and concerns that can help are listed below:

1.      The authors chose 10 or 90 ug/kg dose for BPAF exposure, is there a logical reasoning behind choosing these doses? If so, it would be good to discuss this in the introduction section.

2.      In Figure 1 E the authors claim that the epithelium height is increased after treatment with BPAF, however data suggests that there is no change with BPAF exposure, however treatment with PDTC reduces the height. This needs to be correctly reported.

3.      The findings reported in 1F are the most significant findings of this report and need to be evaluated in further detail. If this data can be backed up by additional experiments like quantitative PCR that would be a great addition to the article. 

4.      Similar to the quantification of IHC in Figure 1 the authors must quantify the results from the IHC performed in Figure 2 and perform statistics on that data.

5.      The authors did not describe the data presented in figure 3 panels A through D in the results section. This data needs to be discussed.

6.      Moreover, the results from ELISA show significance only when quantifying p65 and EGFR, but not with COX2 and TNF-a. Throughout the results, discussion and conclusion section the authors must be careful to report this data as these cannot be reported as an increase in expression. Additional studies such as quantitative PCR can also be carried out to confirm this data.

Author Response

Dear reviewer:

On behalf of all the contributing authors, I would like to express our sincere appreciations of your constructive comments concering our article entitled " Bisphenol AF Induces Prostatic Dorsal Lobe Hyperplasia in Rats through Activation of the NF-κB Signaling Pathway" (Manuscript No: ijms-2638939). These comments are all valuable and helpful for improving our article. According to these comments, we have made corresponding modifications to our manuscript to make our results convincing. In this revised version, changes to our manuscript were all highlighted within the document by using red-colored text. Point-by-point responses are listed below this letter.

We are grateful for your reconsideration of our manuscript, and we look forward to receiving comments from the reviewers.

Yours sincerely,

Jianhui Wu

Dear reviewer:

On behalf of all the contributing authors, I would like to express our sincere appreciations of your constructive comments concering our article entitled " Bisphenol AF Induces Prostatic Dorsal Lobe Hyperplasia in Rats through Activation of the NF-κB Signaling Pathway" (Manuscript No: ijms-2638939). These comments are all valuable and helpful for improving our article. According to these comments, we have made corresponding modifications to our manuscript to make our results convincing. In this revised version, changes to our manuscript were all highlighted within the document by using red-colored text. Point-by-point responses are listed below this letter.

We are grateful for your reconsideration of our manuscript, and we look forward to receiving comments from the reviewers.

Yours sincerely,

Jianhui Wu

Reviewer

The authors present their findings in support of the claim that BPAF results in an increase in prostate weight which is regulated via NFkB pathway. They demonstrate moderate increase in prostate organ coefficient following BPAF exposure which gets reverted when combined with treatment with NFkB inhibitor (PDTC). They also report an increase in height of prostate epithelium (not significant) and increase in PCNA post exposure to BPAF. They further supported their claims with IHC and ELISA data, most of which also demonstrated either no significance or moderate changes between the groups of interest. The article can benefit from additional experiments and deeper analysis of current IHC data. Some of the comments and concerns that can help are listed below:

  1. The authors chose 10 or 90 ug/kg dose for BPAF exposure, is there a logical reasoning behind choosing these doses? If so, it would be good to discuss this in the introduction section.

Response 1: Thank you very much for your comments and professional advice. We have carefully considered your suggestions, and we are pleased to report that we have taken them into account by enhancing the description of our dose selection process within the manuscript's introduction, as per your recommendation. These modifications have been meticulously highlighted in red.

It is important to emphasize that our choice of dosage is grounded in internationally recognized standards. Notably, the World Health Organization has estimated the maximum dietary exposure to Bisphenol A (BPA) in adults to be no greater than 4.2 μg/kg/day [1]. Furthermore, the United States Environmental Protection Agency has reported that the toxicity of BPA exposure, even at levels as low as 2 μg/kg/day, remains a subject of debate within the scientific community [2]. In our prior investigation, we employed a rational approach to assess the potential prostate toxicity of BPA by converting the range of equivalent doses, based on human environmental exposure levels, to their rat counterparts. The results of this endeavor revealed that BPA at low doses (10-90 μg/kg) led to a notable increase in the ratio of serum estrogens to androgens, consequently promoting abnormal hyperplasia in both the ventral and dorsal lobes of the prostate in rats [3,4]. Moreover, the exposure to BPA in the range of 0.01-1 nM was found to enhance the proliferation of prostatic epithelial cells in rats, elevate the expression of estrogen receptors ERα and ERβ, and concurrently decrease the expression of androgen receptor AR. These results collectively provide compelling evidence of low-dose BPA-induced hormonal imbalances and its consequential toxicological effects, particularly in the context of benign prostatic hyperplasia.

In a parallel line of inquiry, we are addressing the analogous compound, Bisphenol AF (BPAF). While it is recognized that BPAF exhibits similar toxicological effects to BPA, the long-term cumulative impact of BPAF on the prostate remains to be definitively established. Therefore, it is of paramount importance to investigate the toxicological effects of low-dose BPAF on the prostate and illustrate the underlying mechanisms. Building upon the dosage parameters established in our previous research [5], which assessed the effects of 10 μg/kg and 90 μg/kg BPA and BPAF on ventral prostate hyperplasia in rats, the present study is poised to delve deeper into the toxic effects and mechanistic aspects of low-dose BPAF on the dorsal lobe of the prostate in rats. This study, in conjunction with our prior work on BPA, is expected to offer valuable insights into the complex realm of endocrine disruptors and their potential reproductive toxicological implications.

  1. Toxicological and health aspects of bisphenol A: Report of joint FAO/WHO expert meeting 2-5 November 2010 and report of Stakeholder Meeting on bisphenol A 1 November 2010, Ottawa, Canada. (World Health Organization and Food and Agriculture Organization of the United Nations). 2011. https://www.who.int/publications/i/item/toxicological-and-health-aspects-of-bisphenol-a
  2. Bisphenol A Action Plan (U.S. Environmental Protection Agency). 2010. https://www.epa.gov/assessing-and-managing-chemicals-under-tsca/bisphenol-bpa-summary
  3. Huang, D.Y.; Zheng, C.C.; Pan, Q.; Wu, S.S.; Su, X.; Li, L.; Wu, J.H.; Sun, Z.Y. Oral exposure of low-dose bisphenol A promotes proliferation of dorsolateral prostate and induces epithelial-mesenchymal transition in aged rats. Sci Rep 2018, 8, 490, doi:10.1038/s41598-017-18869-8.
  4. Wu, J.; Huang, D.; Su, X.; Yan, H.; Sun, Z. Oral administration of low-dose bisphenol A promotes proliferation of ventral prostate and upregulates prostaglandin D(2) synthase expression in adult rats. Toxicol Ind Health 2016, 32, 1848-1858, doi:10.1177/0748233715590758.
  5. Wang, K.; Huang, D.; Zhou, P.; Su, X.; Yang, R.; Shao, C.; Ma, A.; Wu, J. Individual and Combined Effect of Bisphenol A and Bisphenol AF on Prostate Cell Proliferation through NF-kappaB Signaling Pathway. Int J Mol Sci 2022, 23, doi:10.3390/ijms232012283.

  1. In Figure 1 E the authors claim that the epithelium height is increased after treatment with BPAF, however data suggests that there is no change with BPAF exposure, however treatment with PDTC reduces the height. This needs to be correctly reported.

Response 2: Thank you very much for your guidance and valuable suggestions. We have changed "significantly increased " to "an increasing trend" in discussion of the article, and the relevant changes, including results part and discussion part, have been highlighted in red. For more details, we have concluded below:

Results part: " Moreover, the measurement of the epithelial height of DLP in rats (Fig 1E) showed that 10 and 90 μg/kg BPAF could up-regulate the height of DLP." instead of " Moreover, the measurement of the epithelial height of DLP in rats (Fig 1E) showed that 10 and 90 μg/kg BPAF demonstrated a propensity to elevate the epithelial height of DLP. ".

Discussion part: " containing a substantial increase in the epithelial height of DLP " was replaced by " containing an increasing trend in the epithelial height of DLP".

  1. The findings reported in 1F are the most significant findings of this report and need to be evaluated in further detail. If this data can be backed up by additional experiments like quantitative PCR that would be a great addition to the article.

Response 3: We extend our sincere appreciation to the reviewers for your valuable insights and thoughtful recommendations. In response to the inquiry regarding the inclusion of quantitative PCR (qPCR) experiments, we wish to acknowledge that this was indeed considered during the formulation of our experimental design. We concur with your comments that the incorporation of qPCR analysis would facilitate a more comprehensive elucidation of the positive expression rate of PCNA. Regrettably, due to temporal and resource constraints, as well as the limited animal samples, we encountered limitations in the availability of tissue specimens required to undertake an exhaustive analysis including PCNA through IHC and qPCR methods.

In aligning our approach with established experimental protocols found in authoritative journals [1,2], and drawing upon the preceding investigations conducted by our research group [3], we conscientiously opted for the IHC assay as a means to gauge the extent of positive PCNA expression at the protein level. While we acknowledge that the integration of qPCR to analyze PCNA mRNA levels would undoubtedly offer a more comprehensive understanding of the cellular dynamics, we are compelled to defer such analysis to future research endeavors. We wholeheartedly appreciate the reviewers' discerning input, which has contributed to the enhancement of our study.

  1. Zhang, W.; Wang, L.; Sun, X.H.; Liu, X.; Xiao, Y.; Zhang, J.; Wang, T.; Chen, H.; Zhan, Y.Q.; Yu, M.; et al. Toll-like receptor 5-mediated signaling enhances liver regeneration in mice. Mil Med Res 2021, 8, 16, doi:10.1186/s40779-021-00309-4.
  2. Hong, G.L.; Park, S.R.; Jung, D.Y.; Karunasagara, S.; Lee, K.P.; Koh, E.J.; Cho, K.; Park, S.S.; Jung, J.Y. The therapeutic effects of Stauntonia hexaphylla in benign prostate hyperplasia are mediated by the regulation of androgen receptors and 5alpha-reductase type 2. J Ethnopharmacol 2020, 250, 112446, doi:10.1016/j.jep.2019.112446.
  3. Wang, K.; Huang, D.; Zhou, P.; Su, X.; Yang, R.; Shao, C.; Ma, A.; Wu, J. Individual and Combined Effect of Bisphenol A and Bisphenol AF on Prostate Cell Proliferation through NF-kappaB Signaling Pathway. Int J Mol Sci 2022, 23, doi:10.3390/ijms232012283.

  1. Similar to the quantification of IHC in Figure 1 the authors must quantify the results from the IHC performed in Figure 2 and perform statistics on that data.

Response 4: Thank you very much for your attention and valuable suggestions. In order to facilitate typesetting and inter-group comparison with the ELISA data results in figure 3, our quantitative analysis of IHC data is in Figure 3A-D, but we regret to acknowledge an oversight in the earlier version of the manuscript where we failed to provide an illustrative representation for this specific dataset. We recognize that this omission hinders the comprehensive understanding of our research outcomes. To rectify this deficiency, we have now addressed this issue by inserting the illustration and the relevant changes have been highlighted in red. For more details, we have concluded below:

The results of semi-quantitative analysis have revealed that BPAF could signifi-cantly enhance the positive expression of NF-κB p65, COX-2 and EGFR in DLP (Fig 3A, B, D), but demonstrated no significant effect on the expression of TNF-α (Fig 3C). Furthermore, 90 μg/kg BPAF combined with inhibitor PDTC exhibited a notable reduction in the positive ex-pression of NF-κB p65, COX-2 and EGFR in DLP. However, the mitigating influence of PDTC on the expression of TNF-α was not deemed statistically significant.

  1. The authors did not describe the data presented in figure 3 panels A through D in the results section. This data needs to be discussed.

Response 5: Thank you very much for your attention and valuable suggestions. We acknowledge that an illustration was not included in our initial manuscript when describing this result, and the relevant supplement has been highlighted in red. For more details, we have concluded below:

The results of semi-quantitative analysis have revealed that BPAF could signifi-cantly enhance the positive expression of NF-κB p65, COX-2 and EGFR in DLP (Fig 3A, B, D), but demonstrated no significant effect on the expression of TNF-α (Fig 3C). Furthermore, 90 μg/kg BPAF combined with inhibitor PDTC exhibited a notable reduction in the positive ex-pression of NF-κB p65, COX-2 and EGFR in DLP. However, the mitigating influence of PDTC on the expression of TNF-α was not deemed statistically significant.

  1. Moreover, the results from ELISA show significance only when quantifying p65 and EGFR, but not with COX2 and TNF-a. Throughout the results, discussion and conclusion section the authors must be careful to report this data as these cannot be reported as an increase in expression. Additional studies such as quantitative PCR can also be carried out to confirm this data.

Response 6: Thank you very much for your comments and professional advice. We have scrutinized the language employed in the presentation of our results, discussions, and conclusions, ensuring their alignment with the standards of scientific rigor. The relevant modifications have been highlighted in red.

In our investigation, while no statistically significant disparities were observed in the expression levels of COX2 and TNF-α within the dorsal lobe of the rat prostate following exposure to BPAF, it is imperative to note that our prior research findings have conclusively demonstrated the capacity of BPAF to induce the excessive production of COX2 and TNF-α in both the ventral prostate and the human normal prostate stromal immortalized cell line WPMY-1, and the inhibitor PDTC could reduce the expression activity of these indices within the NF-κB signaling pathway.

The difference of index expression observed between the dorsal and ventral lobes of the prostate could be attributed to inherent structural divergences between these two anatomical regions. The ventral and dorsal lobes of the prostate are associated with distinct physiological functions and dissimilar cellular compositions. It is reasonable to posit that these structural differences exert an influence on the response of these respective lobes to the deleterious effects of BPAF exposure. In this context, our findings imply a greater degree of consistency in the expression of relevant indices within the ventral lobe.

It is of particular note that although statistical significance is not achieved, the discernible upward trend in COX2 and TNF-α expression within the dorsal lobe of the prostate subsequent to BPAF exposure, coupled with the concomitant reduction in COX2 and TNF-α expression within the dorsal lobe following treatment with PDTC, is a noteworthy observation that indicates the potential influence of BPAF and the modulatory effect of PDTC.

In addition, we thank you for your thoughtful suggestions. With respect to the implementation of qPCR within our experimental design, we concur with your comments that its inclusion would enhance our ability to provide a comprehensive illustration of the specific expression patterns pertaining to the constituents of the NF-κB signaling pathway. It is indeed a point well taken. Regrettably, we must concede that due to inherent constraints pertaining to time and resource availability, as well as the nature of our experimental material being derived from animal tissues rather than cell lines, we encountered limitations in terms of acquiring a sufficient quantity of tissue samples to undertake an exhaustive array of experiments. Specifically, this pertains to the feasibility of conducting experiments encompassing histopathological assessments, IHC, ELISA.

In light of these constraints, we judiciously opted to focus our efforts on IHC and ELISA analyses, which are deemed to be two robust experimental approaches at the protein level. These selections were made with the intention of providing a more comprehensive understanding of the underlying mechanisms. It is important to emphasize that this approach is in line with the objectives and methodologies previously employed in our investigation of the role of BPAF in the ventral prostate of rats, as documented in our prior research.

We express our gratitude once again for the invaluable guidance and feedback provided by the reviewers, and we remain open to the prospect of further exploring the proposed avenue of inquiry in subsequent research endeavors.

Round 2

Reviewer 2 Report

Comments and Suggestions for Authors

The authors have modified the manuscript according to the suggestions and concerns raised by the reviewers. This has improved the manuscript significantly. I am satisfied with the current version and would recommend the acceptance of this submission.